# A meta-evaluation of the quality of reporting and execution in ecological meta-analyses

Paula Pappalardo[1]*, Chao Song[2], Bruce A. Hungate[3], Craig W. Osenberg[4]

1 Smithsonian Environmental Research Center, Tiburon, California, United States of America, 2 State Key Laboratory of Herbage Improvement and Grassland Agro-ecosystems and College of Ecology, Lanzhou University, Lanzhou, Gansu, China, 3 Department of Biological Sciences, Northern Arizona University, Flagstaff, Arizona, United States of America, 4 Odum School of Ecology, University of Georgia, Athens, Georgia, United States of America

* paulapappalardo@gmail.com

## Abstract

Quantitatively summarizing results from a collection of primary studies with meta-analysis can help answer ecological questions and identify knowledge gaps. The accuracy of the answers depends on the quality of the meta-analysis. We reviewed the literature assessing the quality of ecological meta-analyses to evaluate current practices and highlight areas that need improvement. From each of the 18 review papers that evaluated the quality of meta-analyses, we calculated the percentage of meta-analyses that met criteria related to specific steps taken in the meta-analysis process (i.e., execution) and the clarity with which those steps were articulated (i.e., reporting). We also re-evaluated all the meta-analyses available from Pappalardo et al. [1] to extract new information on ten additional criteria and to assess how the meta-analyses recognized and addressed non-independence. In general, we observed better performance for criteria related to reporting than for criteria related to execution; however, there was a wide variation among criteria and meta-analyses. Meta-analyses had low compliance with regard to correcting for phylogenetic non-independence, exploring temporal trends in effect sizes, and conducting a multifactorial analysis of moderators (i.e., explanatory variables). In addition, although most meta-analyses included multiple effect sizes per study, only 66% acknowledged some type of non-independence. The types of non-independence reported were most often related to the design of the original experiment (e.g., the use of a shared control) than to other sources (e.g., phylogeny). We suggest that providing specific training and encouraging authors to follow the PRISMA EcoEvo checklist recently developed by O'Dea et al. [2] can improve the quality of ecological meta-analyses.

## Introduction

Meta-analyses evaluate summary statistics from primary studies to obtain aggregate effects, assess the heterogeneity of those effects, and ascertain possible causes of the observed heterogeneity. For example, meta-analysis has been used to quantify the strength of density-dependence [3], to assess the response of ecosystems to climate change [4], and to evaluate the

**Data Availability Statement:** All the data files and R code used to analyze and report data are provided in the Supporting information.

**Funding:** During the development of this project CWO received support from the National Science Foundation (DEB-1655426 and OCE-1851032); BH, PP and CWO were supported by a grant from the US Department of Energy, Office of Science, Biological and Environmental Research Program (DE-SC-0010632). The funders had no role in study design, data collection and analysis, decision to publish, or preparation of the manuscript.

**Competing interests:** The authors have declared that no competing interests exist.

performance of different management strategies [5]. Through synthesis, meta-analysis not only advances basic ecological theory, but also facilitates the application of ecological data to inform environmental policy [6]. Moreover, meta-analysis can help identify knowledge gaps, and thus direct new research endeavors [7]. Along with these benefits, the number of published meta-analyses is rapidly increasing [8,9], due to increased data availability, and pressing ecological questions that require synthetic research.

Despite their importance and wide application, the quality of meta-analyses is highly variable [1,7,10,11]. If the quality of meta-analyses is poor, it is hard to know if "biological meta-analysis embodies 'mega-enlightenment', a 'mega-mistake', or something in between" [12]. One issue that can prevent readers from evaluating the overall quality of a published meta-analysis is the lack of details describing each step in the meta-analysis. We refer to this as reporting quality, which is the degree to which the meta-analysis explicitly reports the steps taken to conduct the meta-analysis, including details about the methods used to identify studies and extract data, the meta-analytic model, the number of effect sizes, and the sources of non-independence [7]. Good reporting quality means also that the meta-analysis provides the data used and describes each step of analysis in sufficient detail to replicate the results. Poor reporting quality hinders the readers from assessing if the meta-analysis was executed properly and if the results are reliable. Reporting quality it does not refer to whether those steps were the best available–only that the author(s) was explicit about the steps taken.

The quality of a meta-analysis also is affected by how well the study is implemented. We refer to this proper implementation as execution quality, which is the extent to which the analyses conform to expert recommendation. Examples of recommended execution steps are weighing effect sizes by study precision, testing for publication bias, quantifying heterogeneity in effect sizes, exploring temporal changes in effect size, controlling for phylogenetic non-independence (if applicable), and conducting sensitivity analyses [7].

New methodological guidelines specifically designed for ecology and evolutionary biology [PRISMA-EcoEvo, 2] provide authors, reviewers, and editors with a checklist of items with the goal of improving the overall quality of ecological meta-analysis. Wide adoption of these guidelines could greatly improve the quality of meta-analyses in ecology and evolutionary biology. Assessing the current compliance with recommended steps for reporting and execution in ecological meta-analyses and identifying places that need improvements can help guide the meta-analytic community towards more robust inference and reduce controversy.

In this paper, we reviewed the literature assessing the quality of ecological meta-analyses, collected new data to evaluate current practices, and highlight the areas that need more work. First, we compiled information from 18 studies in the last 20 years (between 2002 and 2022) that reviewed the quality of meta-analyses in ecology, evolution and related fields. These papers provided different insights on the compliance with different standards of reporting quality and with recommended execution steps that should be part of a meta-analysis. Second, we evaluated the recognition and treatment of non-independence for the ecological meta-analyses included in Pappalardo et al. [1], and extracted new data on quality criteria to compare it with the other reviews. Finally, we summarized the level of compliance for different quality criteria across these 18 previous meta-analysis reviews and the new data taken from papers reviewed in Pappalardo et al [1].

## Methods

### Literature search

To evaluate current practices when conducting and reporting ecological meta-analyses, we surveyed the literature for quantitative assessments on criteria previously identified as best

practices in meta-analysis. These criteria fall into two broad categories: 1) execution (i.e., methodological issues related with best practices for data analysis), and 2) reporting (i.e., details describing each step of the meta-analysis and providing the data and information needed to allow for reproducibility). Both categories of criteria aim to ensure appropriate and reproducible results. Our list of criteria was informed by Koricheva & Gurevitch [7, Table 3] and the PRISMA EcoEvo checklist [2]. To find relevant papers, we first performed an exploratory search in Google Scholar, using combinations of keywords including "meta-analysis", "review", "quality", "ecology", "evolution". We then searched the Core Collection of the ISI Web of Science database including articles and reviews within the "Ecology", "Evolutionary Biology", "Biodiversity Conservation" and "Plant Sciences" categories (last search update on Sep 16, 2022). We used a search string for TOPIC as: (["meta-analyses" OR "metaanalyses" OR "meta analyses"] AND ["quality" OR "performance criteria"] AND ["reporting"]). The search resulted in 751 citations. We supplemented those with 11 articles published in the "Meta-analytic insights into evolutionary ecology" Special Issue of Evolutionary Ecology (2012, Volume 26, Issue 5), and 3 articles referenced by other articles or presented in scientific talks. The 765 papers that were obtained were then screened using the *metagear* [13] R package (additional details and R code in S1 Appendix) and based on titles and abstracts, this set was reduced to 61 papers. PDFs of the 61 papers were obtained and evaluated in more detail.

Of the 61 papers for which we screened the full text, the majority were excluded for not having quantitative data on the selected criteria (Fig 1). One paper was excluded for being out of scope. Two of the papers [14,15] overlapped considerably by evaluating many of the same studies in restoration ecology (S2 Appendix); Romanelli et al. [14] describe "many of the materials related to the dataset and methods used to collate evidence are similar to those presented in [15]". We checked their list of references and 92 citations were shared. To reduce non-

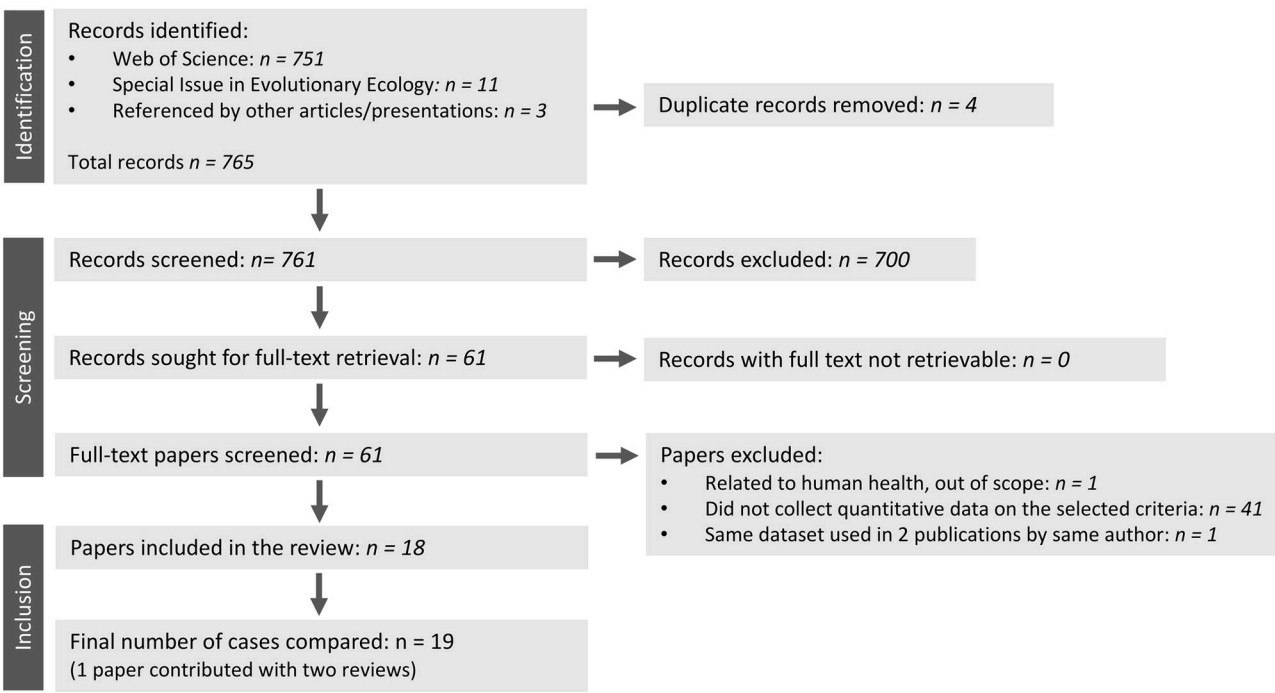

**Fig 1. PRISMA diagram.** The PRISMA diagram details the relevant literature sources identified, screened, excluded, assessed, and selected for the final analysis.

**Table 1. Compilation of 18 papers that reviewed the quality of reporting in ecological meta-analyses.**

| Publication | Review ID | Area | Time period | Number of meta-analysis papers reviewed |
|---|---|---|---|---|
| Archmiller et al. [18] | arch2015 | molecular ecology | 2003–2014 | 18 |
| Beillouin et al. [19] | beil2022 | biodiversity conservation, ecology; environmental sciences | 2001–2020 | 217[a] |
| Cadotte et al. [8] | cado2012 | ecology | 1992–2008 | 240 |
| Chamberlain et al. [20] | cham2012 | ecology and evolutionary biology | 1992–2010 | 56 |
| Chaudhary et al. [21] | chau2010 | ecology | 1992–2006 | 188 |
| Gates [10] | gate2002 | ecology | 1992–1998 | 29 |
| Jennions et al. [22] | jenn2012 | sexual selection | 1996–2012 | 94 |
| Koricheva & Gurevitch [7] | kori2014 | plant ecology | 1996–2013 | 322 |
| Lodi et al. [16] | lodi2021_fe | freshwater ecology | 1994–2017 | 114 |
| | lodi2021_ee | ecology and evolution | 1992–2014 | 86[b] |
| Nakagawa & Santos [23] | naka2012 | ecology and evolution | 2009–2011 | 100 |
| Nakagawa et al. [24][c] | naka2022 | ecology and evolutionary biology | 2010–2019 | 102 |
| Odea et al. [2] | odea2021 | ecology and evolutionary biology | 2010–2019 | 102 |
| Pappalardo et al. [1] | papp2020 | ecology, climate change | 2013–2016 | 96 |
| Philibert et al. [25] | phil2012 | agronomy | 2001–2011 | 73[d] |
| Roberts et al. [26] | robe2006 | conservation, ecology, and environmental management | 2003–2005 | 73 |
| Romanelli et al. [15] | roma2021a | restoration ecology | 2009–2019 | 63 |
| Senior et al. [27] | seni2016 | ecology and evolution | 1992–2014 | 325 |
| Vetter et al. [28] | vett2013 | ecology and conservation biology | 2002–2011 | 133 and 83[e] |

In this table we provide details for each review paper included in our final analysis. The "Review ID" (first author initials plus publication year) was used to identify review papers in Figs 1 and 2, and tables and figures in the Supporting Information. "Area" indicates the subdiscipline(s) summarized in the review papers. Because Lodi et al. [16] reviewed the quality of meta-analyses in two topic areas, we distinguished them using the "Review ID" (lodi2021_fe and lodi2021_ee). Because the two Romanelli et al. reviews [14,15] were based on a similar set of meta-analyses, we included only roma2021a [16] in our final dataset. "Time period" gives the range of publication dates of the meta-analyses that were reviewed.

[a] For Beillouin et al. [19], we counted the number of meta-analyses provided in the supplementary data table referred to as "retained meta-analyses" (which yielded 217 meta-analyses), even though the main text referred to 196 meta-analyses. In addition, most of the percentages mentioned in the main text agreed with the total being 217, rather than 196.

[b] Lodi et al. [16] reviewed n = 86 papers from the n = 325 papers in Senior et al. [27] for their subset of meta-analysis in ecology and evolution (they used the papers that Senior et al. [27] included in their second order meta-analysis). To avoid non-independence between these two review papers, we only collected data from Lodi et al. [16] for criteria that were not addressed by Senior et al. [27].

[c] Nakagawa et al. [24] reviewed the compilation of papers in O'Dea et al. [2] extracting additional information for our criteria of interest.

[d] Philibert et al. [25] analyzed 73 meta-analyses from 55 papers.

[e] Vetter et al. [28] reviewed 133 papers, from which they quantified the percentage that did not report using weights. Then, from the 83 papers that used weights, they quantified the percentage of papers that quantified and explored heterogeneity.

independence in our compilation, we kept only [15] that had information for six of our selected criteria; i.e., we excluded [14] from our analyses. We compiled information from the 18 papers (Table 1) that provided quantitative data on the quality of reporting or execution of the meta-analyses. Because Lodi et al. [16] provided metrics for two separate reviews in two topic areas, our final analyses were based on 19 cases, taken from 18 papers (Fig 1). A PRISMA plot [17] is presented in Fig 1, which details the number of papers in each screening step.

## Overlap between review papers

If there is overlap in the meta-analyses evaluated in these review papers, comparisons between review papers may not be independent; however, because each review used a different set of

search algorithms and often targeted a specific topic, such overlap might be small. We quantified the overlap between meta-analysis reviews for all cases in which the full list of references was available (in the main text or in the supplementary material), or when the authors replied to our requests for this information (S2 Appendix). We used the first author's last name, journal, and year as the identification string to measure overlap in the number of publications shared between review papers.

## Compliance with reporting and execution criteria

Our final list of criteria for analysis is presented in Table 2, where we detail which paper contributed data to each criterion. Because different review papers used slightly different criteria (or different names to refer to similar criteria), we matched similar criteria and provided details on which information was extracted for each review paper in S3 Appendix. We obtained the proportion of the meta-analyses that complied with a particular criterion (data are available in the supplementary data file "compilation-of-previous-review-papers"). For the final analysis, we included only those criteria for which we could gather information from at least two review papers.

In addition, because we had access to the full set of meta-analyses reviewed by Pappalardo et al. [1], we expanded on their results by adding additional criteria detailed below and highlighted in Table 2 and S3 Appendix. Pappalardo et al. [1] analyzed 96 meta-analyses related to global change (PRISMA diagram available in their S1 Fig). For the criteria related to Reporting, we collected new information on: inclusion/exclusion criteria, the number of papers and the number of effect size estimates, the types of non-independence, and if the software, specific functions, and code used for the analyses were provided (when applicable). For the criteria related to Execution, we compiled new data to evaluate if the publication explored temporal changes in effects, conducted sensitivity analyses, controlled for phylogenetic non-independence, and tested for publication bias. More details on the calculations for each criterion are provided in S3 Appendix.

We compiled the percentage of meta-analyses in each review paper that complied with each Reporting and Execution criterion. We classified performance for each criterion as "high" when the percentage of papers complying with a criterion was $\geq 75\%$, "moderate" when compliance was $\geq 50\%$ but $<75\%$, "low" when compliance was $\geq 25\%$ but $<50\%$, and "very low" when compliance was $<25\%$.

## Non-independence

A portion of the new data we collected from the meta-analyses reviewed by Pappalardo et al. [1] focused on non-independence. When a publication acknowledged non-independence (e.g., described some type of non–independence), we also recorded the source of the non-independence that was acknowledged, if the authors attempted to account for it, and the methods used to address non-independence. Non-independence arising from non-independent within-study error (e.g., multiple measurements of the same individual, shared control or treatments) was coded as "sample", whereas non-independence arising from study-level correlation (e.g., multiple effect sizes from each publication which could generate random paper effects) was coded as "study". If a study reported both sources of non-independence, we recorded both (e.g., coding the study as "sample, study"). To code if a publication addressed non-independence, we used: "yes", when the publication described one or more sources of non-independence and addressed at least one; and "no" when the publication did not address non-independence. We coded the methods used to address non-independence as: 1) "average", when the non-independent values were averaged (e.g., [29] averaged repeated measurements,

**Table 2. List of reporting and execution criteria compiled from reviews of meta-analyses.**

| Criteria | Definition | Reviews with data available |
|---|---|---|
| *Reporting* | | |
| Full details of bibliographic searches | Described details such as the databases searched, if specific filters were used, the key words used, and the time span of the review. | [1,2,7,10,15,16,18,19,25,26] |
| Inclusion/exclusion criteria | Clearly described the process of screening and study selection, detailing the criteria used to include (or exclude) studies. | [1 new data,2,7,10,15,16,18,19,26] |
| Reference list of primary studies | Provided the full citation for all the primary studies included in the meta-analysis. | [1,2,7,10,16,18,25,26] |
| Meta-analytical model | Explained the type of meta-analytic model used to analyze the effect sizes (e.g., a random-effects model) and the type of factors and model structure for more complex models. | [1,2,7,16,18,19,27] |
| Dataset used in the meta-analysis | Provided the data used for the meta-analysis: the effect sizes and their variances (when applicable) and moderators (i.e., moderators), if used. | [1,2,7,16,18,25] |
| Data used to calculate effect sizes (raw data) | Provided the data used to calculate effect sizes (e.g., the mean and number of replicates for treatment and control from each comparison). | [1,15,19] |
| The number of papers and the number of effect size estimates | Provided the final number of papers included in the literature review and the number of effect size estimates included in the meta-analysis. | [1 new data,2,8] |
| The software used | Identified the software used to conduct the meta-analysis. | [1,2,7,16,18,23,25,27] |
| The packages used (if applicable) | Identified the packages used to conduct the meta-analysis, if applicable (e.g., if a scripting program like R was used and *metafor* package was used). | [1 new data,2] |
| The functions used (if applicable) | Identified the functions used for data analysis, if applicable (e.g., the rma.mv function from the *metafor* R package). | [1 new data,2] |
| The code used (if applicable) | Provided the code used to conduct the meta-analysis, if applicable (e.g., if a scripting program like R was). If the full code is provided, packages and functions are available. | [1 new data,2] |
| The types of non-independence | Described the sources of non-independence. For example, a non-independent within-study error could occur when there are multiple measurements of the same individual or shared control or treatments; and non-independence could also emerge from study-level correlation (e.g., when there are multiple effect sizes from each publication which could generate random paper effects). | [1 new data,2,16,18] |
| *Execution* | | |
| Weighted effect sizes by study precision | The meta-analysis weighted effect sizes by study precision. The most used weight is the inverse of the variance, but weights can also be based on sample size. | [1,7,15,16,18,19,25,28] |
| Quantified heterogeneity in effect sizes | The meta-analysis provided heterogeneity statistics (e.g., Q statistics, $I^2$, $\tau^2$). | [2,7,10,15,16,18,19,25–28] |
| Explored causes of heterogeneity | The causes of heterogeneity were explored using explanatory variables either through statistical analyses or graphical visualizations. | [1,2,7,10,16,18,19,21,27,28] |
| Conducted multifactorial analysis of moderators | When multiple moderators were included, the non-independence among moderators was accounted for by including all the moderators in the same model. | [7,16,21,27] |
| Tested for publication bias | Publication bias was assessed with any of the recommended methods (e.g., funnel plots). | [1 new data,2,7,10,15,16,18,19,23,25,26] |
| Conducted sensitivity analysis | Quantified the effect of different methodological choices by conducting a sensitivity analysis: e.g., comparing results of a weighted analysis with a reduced dataset versus an unweighted analysis with the full dataset. | [1 new data,2,7,10,16,18,25,26] |
| Explored temporal changes in effect size | Temporal changes in effect sizes were assessed with any of the recommended methods (e.g., a cumulative meta-analysis). | [1 new data,7,16,24] |
| Controlled for phylogenetic non-independence | When multiple species were included in the meta-analysis, their phylogenetic relatedness was considered. The effect of phylogenetic relatedness can be assessed using a phylogeny, if available, or by using taxonomy as a proxy. | [1 new data,7,8,16,20,22,23] |

In this table we provide broad definitions of Reporting and Execution criteria and detail the meta-analysis reviews from which we extracted information from. Each review may have defined the criteria slightly differently; in S3 Appendix we detail how data was extracted and matched for each criterion and for each review paper. For Pappalardo et al. [1] we highlighted the cases in which we re-reviewed the meta-analyses to compile new data. The list of criteria and definitions were informed by Koricheva & Gurevitch [7, Table 3] and the PRISMA EcoEvo checklist [2, Table 1]. In the Discussion section, we expand on best practices for each issue and available solutions.

[30] averaged across species); 2) "choose", when the authors chose one value from multiple non-independent values (e.g., [31] used last sampling point, [32] used one response variable per study); and 3) "model", when the authors accounted for non-independence within the meta-analytic model (e.g., [33] included paper ID as a random effect, [34] included variance covariance matrix obtained from phylogenetic distances); and 4) "tested", when non-independence was assessed, found not to be demonstrable, and was subsequently ignored (e.g., [35–37]). If a test was done and non-independence was supported, then the paper was coded according to the method used to address non-independence, and not as "tested" (e.g., [38]). When multiples methods to address non-independence were used, they were all listed (e.g., "choose, average").

We analyzed and visualized data using the R software [39] and packages *scales* [40], *flextable* [41], *pander* [42], *kableExtra* [43], *readxl* [44], *ggcharts* [45], and *tidyverse* [46]. All the data files and the code used to compile information, analyze data, and create figures and tables, are provided as Supporting Information.

## Results

### Overlap between review papers

The overlap between review papers was generally low. For Reporting criteria, the median number of shared papers was 2 and the mean was 3; for Execution criteria, the median number of shared papers was 2 and the mean was 4.7. In S2 Appendix we include the overlap matrices for each quality criterion that show the number of papers that overlapped between each review paper, and the distribution of the number of shared papers (S1 and S2 Figs). The two instances with largest overlap were: 1) overlap of 77 papers between Koricheva & Gurevitch [7] and Senior et al. [27], representing a 23.9% overlap for the "Meta-analytical model", "The software used", "Quantified heterogeneity in effect sizes", "Tested for publication bias", and "Multifactorial analysis of moderators criteria" criteria; 2) overlap of 74 papers between Koricheva & Gurevitch [7] and Cadotte et al. [8], representing 30.8% overlap for the "Controlled for phylogenetic non-independence" criterion.

### Compliance with reporting and execution criteria

In our compilation of the 19 meta-analysis reviews from 18 papers, we found wide variability in the compliance within and between the different quality criteria. We did not observe any clear differences among different subdisciplines (S3 and S4 Figs, S4 Appendix), nor did we observe any temporal trends in compliance (S5 and S6 Figs, S4 Appendix).

In general, we observed better compliance in Reporting (Fig 2) than Execution (Fig 3). Across reviews, we observed high to moderate compliance with Reporting criteria such as: providing the list of references (Fig 2F), specifying the meta-analytic model (Fig 2D), detailing inclusion/exclusion criteria (Fig 2C), and identifying the packages (Fig 2J) and software (Fig 2K) used. On the other hand, Reporting criteria exhibited very low to moderate compliance in including full details on the literature search (Fig 2B), providing the data used to calculate effect sizes (Fig 2E), and providing the analytic code (Fig 2G) and functions used (Fig 2H).

For the Execution criteria, there was lower compliance with criteria such as conducting sensitivity analyses (Fig 3A), controlling for phylogenetic non-independence (Fig 3B), exploring temporal changes in effect sizes (Fig 3D), conducting a multifactorial analysis of moderators (vs. multiple single factor analyses) (Fig 3E), and testing for publication bias (Fig 3G). In contrast, most papers explored the possible causes of heterogeneity (Fig 3C). For the Execution criteria of weighting effect sizes by study precision (Fig 3H) and quantifying heterogeneity in

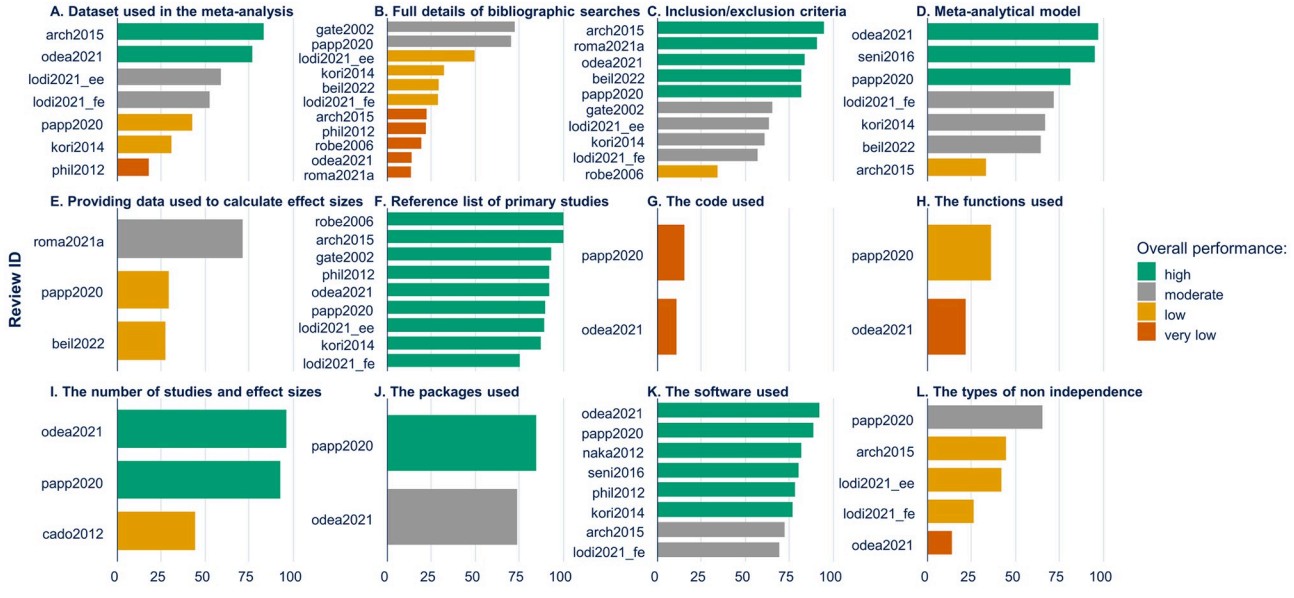

**Fig 2. Percent of papers complying with reporting criteria.** The percent of papers complying with each criterion is plotted for each synthesis paper. The colors indicate the overall performance for each criterion coded as: "High" (percentage compliance ≥ 75%), "moderate" (50 ≤ percentage compliance < 75), "low" (25 ≤ percentage compliance < 50), and "very low" (percent compliance < 25%). The Review ID corresponds to the papers listed in Table 1.

effect sizes (Fig 3F), compliance was highly variable (ranging from 33% to 92% for weighting and 22% to 100% for quantifying heterogeneity).

## Non-independence

In our review of the meta-analyses compiled by Pappalardo et al. [1], we found that in all meta-analyses but one, the number of effect sizes was larger than the number of papers (Fig 4). This variation was often of several orders of magnitude (Fig 4, note the log scale in both axis). This suggests the possibility of non-independence as effect sizes derived from the same source paper are more likely to be more similar than are those coming from different papers. 66% of the meta-analyses acknowledged some type of non-independence (Fig 5A). The source of non-independence acknowledged most often (68% of the time) was related to the design of the original experiment (e.g., a common control used for different treatments) and how data were collected (Fig 5B). Acknowledging non-independence from other sources of correlation (e.g., multiple effect sizes per publication) was less common (36% of the time, Fig 5B). Most papers (98%) that acknowledged non-independence took steps to address it (Fig 5C). The most common ways that non-independence was addressed (Fig 5D) were: choose (55%) and average (32%). Only 11% tested for the effects of non-independence, and only 16% explicitly modeled a potential source of non-independence. A few papers used a combination of these approaches (which is why the percentages sum to slightly more than 100%).

## Discussion

### Compliance with reporting criteria

Even though there was overall good compliance for Reporting criteria (e.g., providing the list of primary papers included in the meta-analysis), many issues remain widespread. Meta-

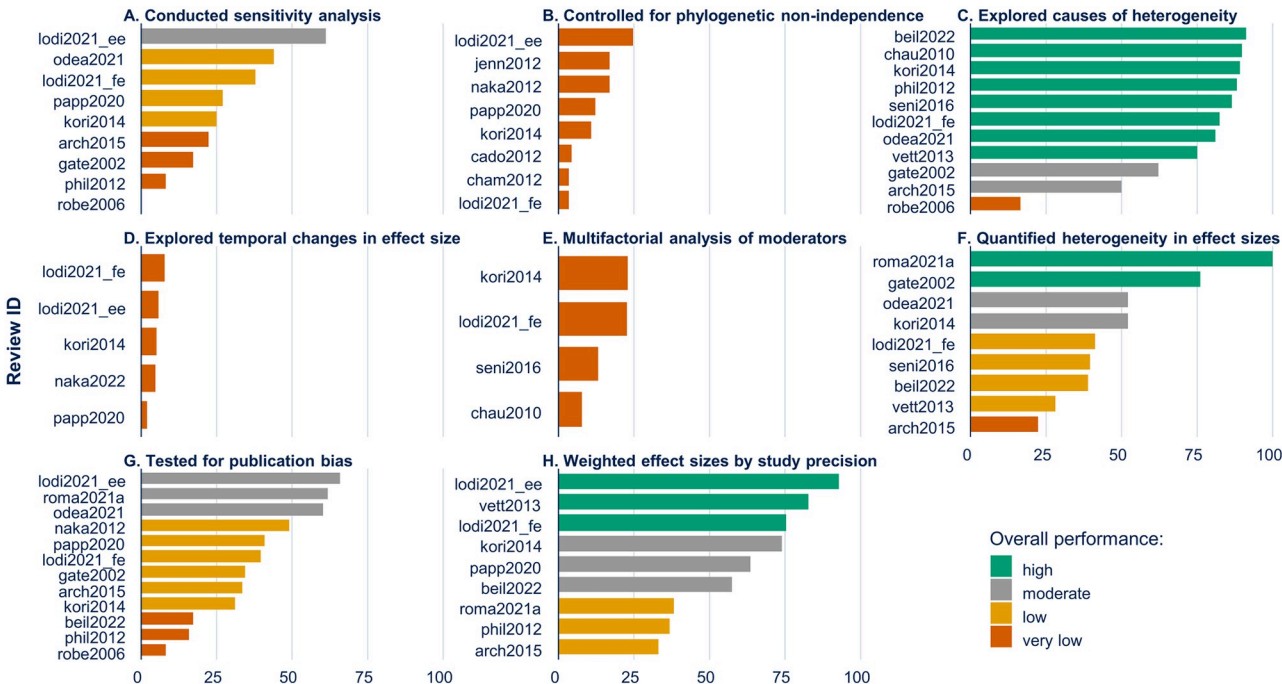

**Fig 3. Percent of papers complying with execution criteria (i.e., recommendations during data analysis).** The percent of papers complying with each criterion is plotted for each synthesis paper. The colors indicate the overall performance for each criterion coded as: "High" (percentage compliance ≥ 75%), "moderate" (50 ≤ percentage compliance < 75), "low" (25 ≤ percentage compliance < 50), and "very low" (percent compliance < 25%). The Review ID corresponds to the papers listed in Table 1. In panel (A), Roberts [26] evaluated sensitivity analysis, and found 0% of papers reporting it.

analysis papers were less consistent in their reporting of information that can critically affect the results of a meta-analysis (e.g., the inclusion/exclusion criteria [49]). Even minimal information such as the number of papers and effect sizes were not always included; for example, the review by Cadotte et al. [8] showed that fewer than 50% of the meta-analyses reported this basic information. Many of these meta-analyses are not reproducible because relatively few of the meta-analyses provided the data used to conduct their analyses (e.g., effect sizes, variances, moderators). Making the data available benefits the research community by supporting meta-research or integrative research that combines the data in some novel way without having to redo the data extraction [50]. Many studies also failed to specify the model used to analyze the data (e.g., random-effects model). To remedy this issue, we suggest academic journals adopt standard checklists for reporting items, such as the PRISMA-EcoEvo checklist [Tables 1 and 2]. Similarly, for the meta-analyses that reported using a programming language, very few reported the specific functions or the code used for data analysis, which are essential for reproducibility. Failure to share code is not exclusive to meta-analysis; even for research articles published in ecological journals that encourage or mandate code-sharing, only 27% provide all or some of the code used for the analyses [51].

To encourage code and data sharing, journals can develop incentives. Some cover the fee for publishing data in a repository. Discounts on open access fees could further encourage authors to share code and data. Most data repositories provide a separate DOI for the dataset so it can be properly cited. Most importantly, as reporting practices improve and data become

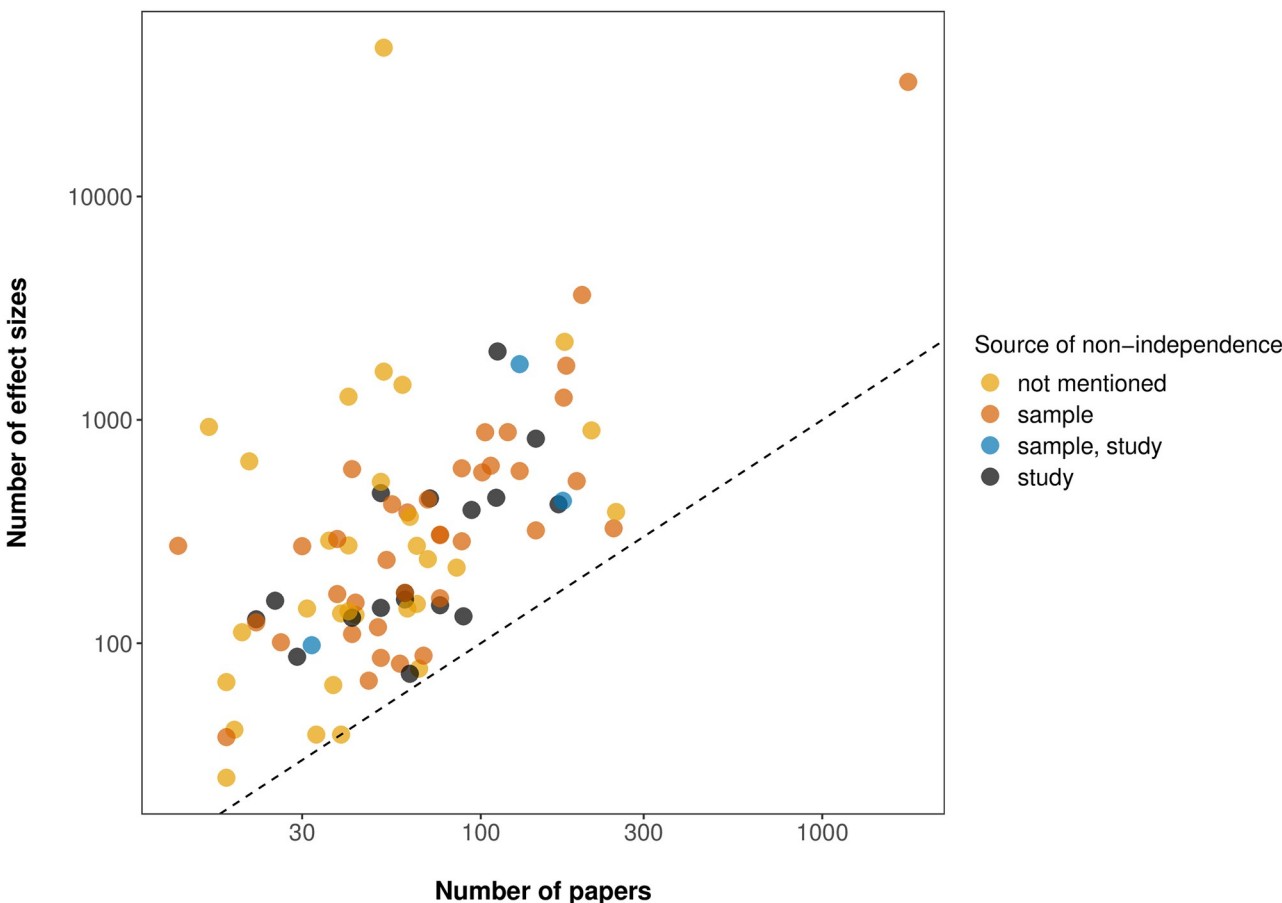

**Fig 4. Relationship between the number of papers and the number of effect sizes included in ecological meta-analysis.** The relationship between the number of papers and the number of effect sizes based on our re-analysis of meta-analyses in Pappalardo et al. [1]. The colors indicate when the source of non-independence was not mentioned (yellow) or was acknowledged at the sample level (orange), study level (black) or both (blue). Note that the axes are on log scales to accommodate two studies with an extreme number of effect sizes. One is a study reporting 52 papers that did not specify the number of effect sizes but provided a dataset with 46,347 rows [47]. The other is a study that analyzed 1,785 papers and reported a total of 32,567 effects for one of their meta-analyses [48]. The dashed line is the one-to-one line (x = y).

available, reproducibility will improve. Achieving computational reproducibility will help ensure results are robust, transparent, and credible. This is particularly important for researchers working in applied science and conservation where stakes are high, and transparency can help maintain public trust [52].

Having commonly accepted guidelines for meta-analysis could improve the quality of meta-analyses, although empirical research on this topic often gives mixed results. Even before the PRISMA guidelines were initially developed [in 2009 by 17], systematic reviews in the medical field showed higher reporting quality compared with meta-analyses in ecology [10,26]. This was likely due to the early guidelines for systematic reviews in the medical field using a standard set of methods developed by the Cochrane Collaboration [53] and to ecological studies often being more complex and varied in terms of the types of questions, sources of data, and experimental design. How much the PRISMA guidelines improved the quality of reporting in medical meta-analysis is not clear. Some papers report a moderate increase in reporting quality after the publication of PRISMA guidelines [54], while others report no change [55, only reviewed abstracts]. Two syntheses of medical meta-analyses found that reporting quality improved after journals endorsed and implemented PRISMA guidelines

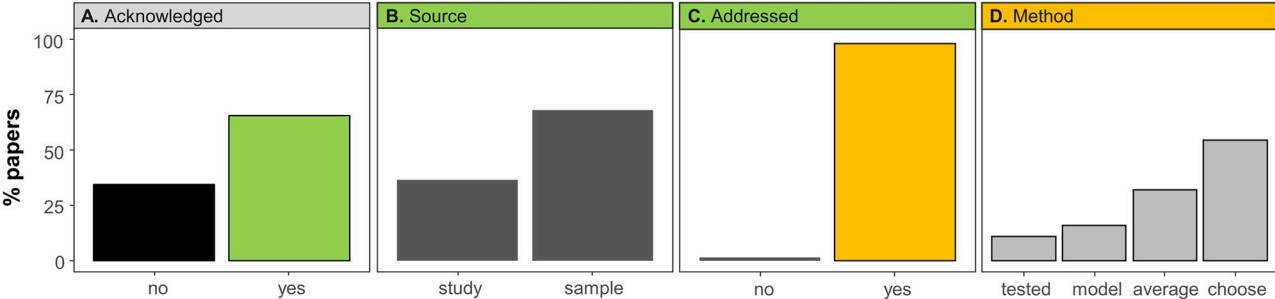

**Fig 5. Percent of papers that acknowledged non-independence, addressed it, and which methods they used to deal with non-independence.** (A) Percent of papers that acknowledged at least one type of independence in their data ("yes") or did not acknowledge non-independence ("no"). (B) For the papers that did acknowledge non-independence, the sources of non-independence were classified as "study" or "sample". (C) The percent of papers that addressed at least one type of non-independence ("yes"), or did not address non-independence ("no"). (D) For the papers that did address non-independence, we show the methods used to address non-independence, classified as: "Choose", when the authors chose one value from multiple non-independent values; "average", when the non-independent values were averaged; "model", when the authors accounted for non-independence within the meta-analytic model, and "tested" when the authors tested for the effects of non-independence. The papers that used more than one method (or source) were counted in each category, so the percentages between levels of panels B and D sum to greater than 100%.

[56,57]. In their review of meta-analyses in ecology and evolution, O'Dea et al. [2] showed that meta-analyses that reported to have followed specific guidelines tended to have higher quality ratings. Lodi et al. [16], in their review of meta-analyses from freshwater ecology, found higher quality in more recent years and suggested that previous papers on reporting guidelines were the reason for the improvement. The recently published PRISMA Eco-Evo guidelines [2] could generate even bigger impacts on the quality of reporting if journals required those guidelines during the submission of meta-analyses. Some journals such as PLOS ONE already have a structure in place to detect if certain key aspects of meta-analysis are present (e.g., a PRISMA plot). We suggest that introducing the PRISMA Eco-Evo guidelines to a journal's submission process will greatly benefit the discipline, especially if the journal publishes a large number of meta-analyses as is the case for Ecology Letters, Global Change Biology, Ecology, Oecologia and American Naturalist (the top five according to our compilation, S5 Appendix).

## Compliance with execution criteria

The low compliance with Execution criteria suggests that most meta-analyses do not follow recommended methods. One of the advantages of meta-analysis is that effect sizes are conventionally weighted by the precision of the observed effect size. The PRISMA-EcoEvo guidelines recommend using a weighted analysis because weighting generally yields more precise estimates of effects that unweighted analyses [2]. However, our compilation of reviews showed that the percentage of papers that weighted effect sizes varied widely (from 33% to 93%). By reanalyzing the meta-analyses from Pappalardo et al. [1], we found that only 42% weighted by the inverse of the variance (as recommended by [2]), 6% weighted by sample size, 16% used some non-traditional weight, and 36% of the meta-analyses did not weight effect sizes in any way to account for variation in their precision or quality. Several papers used unweighted analyses because of incomplete reporting in the primary publications (e.g., the original papers did not report standard deviations or sample sizes), and the meta-analysts did not want to greatly reduce the number of studies by excluding the studies without estimates of variance. New imputation techniques to estimate variances can provide an effective alternative to conducting an unweighted meta-analyses [58]. Under some situations, however, unweighted analyses can provide results as reliable as those obtained using weighted analyses, e.g., when among-study

variance is large relative to within-study variances (Song et al, pers. comm.) or when effect sizes and their variances are independent and follow a normal distribution [58]. Conducting a sensitivity analysis with a smaller dataset that compares results from unweighted and weighted meta-analyses can be a way to check if results are robust to that decision [59].

A central purpose of ecological meta-analysis is to quantify heterogeneity and explore its causes. A fixed-effects model, which assumes no heterogeneity among true effect sizes, has been discouraged for ecological meta-analysis [60] and its use seems to be declining [27]. Pappalardo et al. [1] found that random-effects and mixed-effects models were the most popular in their review. Given that heterogeneity in ecological and evolutionary meta-analyses is high [27], it is encouraging we found high compliance in exploring the causes of heterogeneity (either by conducting statistical analysis of moderators or by graphical visualizations). However, our compilation showed high variability on providing metrics quantifying heterogeneity (e.g., using Q or $I^2$ statistics), with most reviews (seven of nine) reporting very low to moderate compliance on quantifying heterogeneity.

A common approach when meta-analysts explore heterogeneity is to evaluate individual covariates one at a time, rather than in a single analysis. This is an invalid approach because these explanatory variables may not be independent or because failure to simultaneously account for a factor may give rise to spurious results (e.g., via Simpson's paradox). A multifactorial analysis of moderators, which would address this issue, was reported only in a few meta-analyses. Gates [10] also mentioned that most meta-analyses did not correct for multiple testing when conducting subgroup analyses. This deficiency could reflect limitations imposed by available software. For example, MetaWin, a commonly used software in older meta-analyses, did not allow multifactorial analyses. Additionally, Nakagawa & Santos [23] noted that meta-analytic data are often sparse and including all moderators in the model may greatly reduce sample sizes, making such analyses problematic [see 61]. However, given the increase in data availability, and an increase in the sophistication of software capable of including multiple moderators (e.g., the R package *metafor* [62]), multifactorial analysis are more feasible and should become more common.

Sensitivity analyses evaluate the robustness of a meta-analysis to methodological choices, for example, by exploring how results change when removing influential points, altering the weighting schemes, or calculating different types of effect sizes. Sensitivity analyses can also be used to explore the consequences of non-independence [63]. Across reviews, we observed that a low percentage of meta-analyses reported conducting a sensitivity analysis. Although analysis for publication bias can be considered a type of sensitivity analysis, we followed previous reviews (e.g., [7]) and quantified them as separate criteria. It is possible that some researchers may have run additional explorations that could be considered sensitivity analyses in earlier stages of a publication, but these were not stated explicitly in their final manuscripts or supplementary information. We encourage authors of meta-analyses to include the results of sensitivity analyses, to showcase the different types of limitations related to their dataset, and to quantify if different methodological choices affect their conclusions. This will enhance the robustness of their meta-analysis.

A general problem in scientific research is that significant results are more likely to be published. For meta-analysis, this may bias the meta-analyst towards discovering more significant effect sizes, which in turn may bias the conclusion of the meta-analysis. Meta-analysis has tools to identify the existence of publication bias (e.g., via a funnel plot) and to assess its impact (e.g., by calculating a fail-safe number), but these methods have pros and cons, and meta-analysts are discouraged from relying on only one approach [12,23]. Despite the availability of methods, compliance for assessing publication bias was low (<50%) in nine of the twelve reviews that quantified this criterion.

A different type of publication bias in meta-analysis of ecology and evolution arises from temporal trends in effect sizes. For example, a decrease in the magnitude of effect sizes over time has been observed in various areas of ecology and evolution [64,65], although the existence of such a general trend has been debated [66]. Possible non-biological causes for the decrease in the magnitude of effect sizes with time are time-lags, selective reporting, shifts in the choice of research organisms, and changes in statistical methods (reviewed in [65]). Not accounting for temporal trends may give a false sense that conclusions from meta-analyses are invariant through time [67]. Instead, meta-analyses should explore temporal trends, which would also help identify additional sources of heterogeneity in the effect sizes. Koricheva et al. [65] described graphical and statical methods available to analyze temporal trends and included examples analyzing real datasets. The simplest graphical method is a plot of effect sizes versus publication year. Another option is a cumulative meta-analysis, in which mean effect sizes are calculated starting with the oldest publication and adding in other studies chronologically [65]. The temporal effect can also be assessed in a statistical model by incorporating publication year as a moderator [65]. In the ecological meta-analyses reviewed by Pappalardo et al. [1], 40% of meta-analyses reported the range in publication years of the original papers and this time span averaged 41 years (min = 1, median = 34, max = 115). In their review, Cadotte et al. [8] found an average time span of 15 years and a maximum of 65 years. Despite the wide time span reported in many of the meta-analyses reviewed, the percent of meta-analyses that addressed temporal trends in effect size was very low (ranging from 1 to 8%).

Methods for detecting and quantifying the effects of publication bias, such as regression or correlation-based approaches for analyzing the asymmetry of funnel plots, may encounter challenges in ecology and evolution due to heterogeneity and non-independence, two characteristics commonly associated with data in ecology and evolution [24]. To address this issue, Nakagawa et al. [24] proposed using what they referred to as "conditional residuals" from hierarchical models instead of observed effect sizes in analyzing funnel plots. This approach accounts for heterogeneity and non-independence by subtracting the fixed effects and random effects that model the heterogeneity and non-independence from the observed effect sizes.

A Reporting criterion that we could not include in the main analysis was associated with the method used to obtain the uncertainty interval associated with the mean effect size–only one review paper assessed this criterion [1]. However, the method used to obtain an uncertainty interval (e.g., based upon a t-distribution or a z-distribution, or by bootstrapping) can affect the coverage and thus the inferences from the analyses. For example, when replication is low, the use of the bootstrap or z-distribution will generate confidence intervals that are much too narrow, resulting in more significant effects than expected [1]. 39% of the meta-analysis reviewed by Pappalardo et al. [1] did not specify this information, even though it can have dramatic effects on statistical inference. Furthermore, of the meta-analyses that did report their method, a vast majority (>90%) used either the bootstrap or z-distribution [1], which can lead to artificially small confidence intervals.

## Non-independence

Non-independence is common in biological meta-analyses; if not addressed properly, non-independence can produce spurious results [68]. In particular, not properly accounting for non-independence often leads to wrong estimates of standard errors and thus invalid statistical inference. Non-independence may occur at the sampling level, such as through using a shared control or taking repeated measurements of the same individuals over time. Non-independence at the study level may occur by comparing species that are close phylogenetically or systems that are close spatially. Our review of the meta-analyses compiled by

Pappalardo et al. [1] showed a higher percent of papers (66%) that acknowledged some source of non-independence, compared with reviews by Archmiller et al. [18] and O'Dea et al. [2], which reported 44% and 14% respectively. Most non-independence arose at the sampling level, due, for example, to using a shared control or taking repeated measurements on replicates. In these cases, most meta-analysts addressed the non-independence before doing the analysis (one of the solutions mentioned in [63]), by either choosing a subset of the data (55%) or using the average of the non-independent measurements (32%). We did not observe any meta-analyses that tried to explicitly model the covariance of the sample-level non-independence even though formulae for covariance have been derived for many forms of non-independence [69,70].

In contrast, non-independence arising from study-level correlation are much less recognized and addressed in meta-analyses. Only 14% of the meta-analyses from Pappalardo et al. [1] attempted to address study-level non-independence by applying a multilevel model (e.g., using study ID as a random effect in the meta-analytic model). Including study effects in a multilevel model is one the simplest solutions, and different levels can be included in the model to account for non-independence due to other sources (e.g., species effects, discussed below) [12]. Study-level non-independence may arise in multiple ways in ecological meta-analyses [63]. In fact, we found that the number of effects far exceeds the number of papers in the meta-analyses we examined and many of the studies that have a large ratio of number of effect sizes to number of papers did not acknowledge sources of non-independence (Fig 4). Given that studies from the same source paper are more likely to share similar environments or methodology, it is very likely that study-level non-independence is common. Thus, the relatively low proportion of published meta-analyses addressing study-level non-independence is a source of concern.

Study-level non-independence frequently arises from phylogeny. Closely related species may have similar traits that could be associated with similar responses; thus, data from different species may not be independent. In paired analysis of the same dataset using traditional and phylogenetic meta-analyses, Chamberlain et al. [20] reported that 40% of random-effects meta-analyses changed from significant when not adjusting for phylogeny to non-significant when a phylogenetic meta-analysis was used. The influence of phylogenetic relatedness on the outcome of meta-analysis has also been studied using simulations. Cinar et al. [71] found that under moderately strong phylogenetic relatedness, failing to account for species-level variance generated biased estimates of mean effects and led to poor coverage (i.e., confidence intervals that were too small). This is troubling given that all the meta-analysis reviews found very low compliance with respect to controlling for phylogenetic non-independence. In some cases, a phylogenetic analysis may not be possible because a reliable phylogeny is not available. However, in those cases, taxonomic information (e.g., family or genus) can be used as a moderator in the analysis (e.g., as done in [35]).

## How can we implement best practices?

As was highlighted in multiple sessions at the 2020 Ecological Society of America meeting, there is a need for data integration at multiple scales, data synthesis, and training of young investigators on computer programming and the use of appropriate statistical tools. To address this gap, it is important to train ecologists in meta-analysis techniques. This could involve including meta-analysis topics in the curriculum of Ecology/Evolution graduate programs, which could be done as part of courses focused on statical methods and data analysis or the subject could be required for qualifying exams. Training also could be provided in short workshops. The ever-increasing availability of ecological data and the scope of the questions we need to answer, require that we provide all researchers access to the tools necessary for synthesis research.

For researchers who wish to learn on their own, there are multiple resources available. Marc Lajeuneese has a YouTube channel (https://www.youtube.com/c/lajeunesselab) with multiple videos explaining techniques for the different steps to conduct a meta-analysis, and has also developed the R package *metagear* [13] that has functions to help with paper screening and data extraction (http://lajeunesse.myweb.usf.edu/metagear/metagear_basic_vignette.html). The Environmental Computing website (http://environmentalcomputing.net/meta-analysis/) provides tutorials to conduct meta-analysis with the R package *metafor* [62], and also general information on how to organize data that will be useful for meta-analysts (http://environmentalcomputing.net/data-entry/). The *metafor* website by Wolfgang Viechtbauer has detailed documentation and examples of data analysis and models to conduct a meta-analysis using *metafor* (https://www.metafor-project.org/doku.php/metafor). The CRAN task view for meta-analysis [72, https://cran.r-project.org/web/views/MetaAnalysis.html] provides a full list of R packages that have useful tools related with meta-analysis. Also available for R users is the online book *Doing meta-analysis with R*: *a hands-on guide* that is aimed at non-experts [73, https://bookdown.org/MathiasHarrer/Doing_Meta_Analysis_in_R/]. For Python users, there are also specific tools and resources focusing on meta-analysis [74,75]. For those who would prefer a friendly user interface, the software OpenMEE [76] provides advanced tools for meta-analysis in ecology and evolution (http://www.cebm.brown.edu/openmee/help.html). The Inter-Disciplinary Ecology and Evolution Lab (http://www.i-deel.org/links.html) provides several resources related to meta-analysis and systematic reviews. Finally, Briggs et al. [77, http://metaanalysis.ecology.uga.edu/] are developing a series of meta-analysis tutorials.

Now that specific guidelines are available with a focus on meta-analysis in Ecology and Evolution, authors can follow the PRISMA EcoEvo checklist [2] as a guide to plan their meta-analysis, and reviewers and editors can assess the quality of reporting in a meta-analysis. More importantly, improving the reporting quality and following guidelines will also improve the quality of the research. Although we showed that authors are better at following Reporting criteria than Execution criteria, compliance was highly variable suggesting there remains ample room for improvement. Making the PRISMA EcoEvo checklist mandatory for paper submission could help by 1) helping to identify if a paper is not a statistically-focused meta-analysis (e.g., papers that self-report as "meta" analysis because they analyzed a large dataset with multiple factors, but that do not use a meta-analytic framework, and 2) encouraging good reporting, reproducibility, and overall quality. A key component to future meta-analyses and synthesis studies are data sharing and good data management practices [78].

The learning curve to conduct a meta-analysis and follow all the detailed steps may appear steep and discouraging. The training opportunities mentioned above could help reduce the learning curve and facilitate improved reporting and execution. Some researchers argue that we should not let the perfect be the enemy of the good [79]. We agree, but also argue that the "good" should include clearly reporting methods, following best practices, and making data and code available to the community. Doing so will make the inferences from meta-analyses more robust and less controversial—the ultimate goal of a valuable statistical tool.

## Supporting information

**S1 Checklist. PRISMA 2020 checklist.**
(DOCX)

**S1 Fig. Distribution of paper overlap for reporting criteria.** Distribution of the number of papers shared between reviews for all the Reporting criteria combined.
(TIF)

**S2 Fig. Distribution of paper overlap for execution criteria.** Distribution of the number of papers shared between reviews for all the Execution criteria combined.
(TIF)

**S3 Fig. Percent of papers complying with Reporting criteria by review discipline.** The percent of papers complying with each Reporting criterion is plotted for each review paper. The colors indicate different subdisciplines of the review papers. The Review ID corresponds to the papers listed in Table 1 of the main manuscript.
(TIF)

**S4 Fig. Percent of papers complying with Execution criteria by review discipline.** The percent of papers complying with each Execution criterion is plotted for each review paper. The colors indicate different subdisciplines of the review papers. The Review ID corresponds to the papers listed in Table 1 of the main manuscript.
(TIF)

**S5 Fig. Percent of papers complying with Reporting criteria as a function of the time period analyzed by the review paper.** Each panel represents a Reporting criterion. The line segment indicates the time period covered by each of the review papers that addressed a particular criterion.
(TIF)

**S6 Fig. Percent of papers complying with Execution criteria as a function of the time period analyzed by the review paper.** Each panel represents an Execution criterion. The line segment indicates the time period covered by each of the review papers that addressed a particular criterion.
(TIF)

**S1 Appendix. Paper screening.** Additional details and R code for paper screening using the package *metagear*.
(PDF)

**S2 Appendix. Overlap between review papers.** Additional details and R code used to calculate overlap between review papers, including overlap matrices for each criterion and R code for S1 and S2 Figs.
(PDF)

**S3 Appendix. Details on the information extracted from each review paper for each performance criterion.** For data from Pappalardo et al. [1], we indicated when their data was re-analyzed or when we collected new data in this study by re-reviewing their compilation of ecological meta-analyses with the tag "added". When the number of publications complying (or not complying) with one of the criteria was reported, we used that information to calculate the percentage of papers complying; in other cases the reviews directly reported the information as a percentage. In a few papers in which we had the original review data for each criterion [e.g., 18], we summed the number of papers complying with each criterion, and then calculated the percentage of compliance based on the total number of papers relevant for that criterion.
(PDF)

**S4 Appendix. Additional results.** Details and R code for additional results about compliance of quality criteria by review discipline and temporal trends in compliance. Includes the R code to generate S3–S6 Figs.
(PDF)

**S5 Appendix. Journals that publish the most meta-analyses.** Number of meta-analyses per journal that had been included in the meta-analysis reviews. Because the distribution is strongly right skewed (with most journals publishing a few meta-analyses), we display only the journals with at least 5 meta-analyses.
(PDF)

**S6 Appendix. R code used for data analysis.** This rmarkdown file (.Rmd) includes the code to conduct the data analysis and create Figs 1 to 5.
(RMD)

**S1 Data. List of references in reviews.** This Microsoft Excel Worksheet (.xlsx) includes the compilation of all the references analyzed by previous reviews.
(XLSX)

**S2 Data. New data from Pappalardo et al. [1].** This Microsoft Excel Worksheet (.xlsx) includes additional data collected by re reviewing the ecological meta-analysis compiled by Pappalardo et al. [1]. Please cite this publication and Pappalardo et al. [1] if you are using the data in this file for your research.
(CSV)

**S3 Data. Data compilation from previous review papers.** This Microsoft Excel Worksheet (. xlsx) includes information on the quality of Reporting and Execution criteria compiled from the review papers listed in Table 1.
(XLSX)

**S4 Data. Papers screened and final classification.** This Microsoft Excel Worksheet (.xlsx) includes the final list of papers screened using metagear from the Web Of Science search and also the additional papers found from additional sources.
(XLSX)

**S5 Data. Journal names dictionary.** This Comma Separated File (.csv) includes a conversion dictionary from short journal names to long journal names and was used when analyzing paper overlap.
(XLSX)

## Acknowledgments

We are very thankful to Marc Cadotte, Scott Chamberlain, Steve Hovick, Julia Koricheva, Jessica Gurevitch, Shinichi Nakagawa, and João Paulo Romanelli for providing detailed additional methods or metadata associated with their reviews of meta-analyses. We also thank O'Dea et al. [2] for sharing a draft of the PRISMA EcoEvo guidelines before it was published. PP is grateful for the encouragement made by an anonymous reviewer of the Pappalardo et al. [1] publication to expand the analysis on the quality of reporting in meta-analysis in a separate publication. PP highly appreciates the feedback from Alyssa Gehman, Linsey Haram, Carrie Keogh, and Rachel Smith on an early draft of the manuscript.

## Author Contributions

**Conceptualization:** Paula Pappalardo, Chao Song, Bruce A. Hungate, Craig W. Osenberg.

**Data curation:** Paula Pappalardo, Chao Song.

**Formal analysis:** Paula Pappalardo.

**Funding acquisition:** Craig W. Osenberg.

**Investigation:** Paula Pappalardo, Chao Song.

**Methodology:** Paula Pappalardo, Chao Song, Craig W. Osenberg.

**Software:** Paula Pappalardo.

**Validation:** Paula Pappalardo, Chao Song.

**Visualization:** Paula Pappalardo.

**Writing – original draft:** Paula Pappalardo.

**Writing – review & editing:** Paula Pappalardo, Chao Song, Bruce A. Hungate, Craig W. Osenberg.

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
