## [Decision Letter · Decision Letter 0]

13 Jun 2023

PONE-D-22-24320A meta-evaluation of the quality of reporting and execution in ecological meta-analysesPLOS ONE

Dear Dr. Pappalardo,

Thank you for submitting your manuscript to PLOS ONE. After careful consideration, we feel that it has merit but does not fully meet PLOS ONE’s publication criteria as it currently stands. Therefore, we invite you to submit a revised version of the manuscript that addresses the points raised during the review process.

We look forward to receiving your revised manuscript.

Kind regards,

Daniel de Paiva Silva, Ph.D.

Academic Editor

PLOS ONE

Additional Editor Comments:

Dear Dr. Pappalardo,

After this frist review round your manuscript has received very favorable deicisions. All three reviewers recommended a minor review should be implemented before your manuscript is accepted for publication in PLoS One Consider the extent of improvements, I will grant you a one-month period to deliever the new version of your manuscript to PLoS One. Please do not forget to prepare a rebuttal letter informing the authors what were the changes made and justifying those suggestions that were not accpeted by the authors.

Congratulations on your hard work on this MS.

Sincerely,

Daniel Silva

Reviewers' comments:

Reviewer's Responses to Questions

**Comments to the Author**

1. Is the manuscript technically sound, and do the data support the conclusions?

Reviewer #1: Yes

Reviewer #2: Yes

Reviewer #3: Yes

2. Has the statistical analysis been performed appropriately and rigorously? 

Reviewer #1: No

Reviewer #2: Yes

Reviewer #3: Yes

3. Have the authors made all data underlying the findings in their manuscript fully available?

Reviewer #1: Yes

Reviewer #2: Yes

Reviewer #3: Yes

4. Is the manuscript presented in an intelligible fashion and written in standard English?

Reviewer #1: Yes

Reviewer #2: Yes

Reviewer #3: Yes

5. Review Comments to the Author

Reviewer #1: Dear Authors, your article is quite interesting. The topic you chose is captivating, and your insights are thought-provoking.

However, as I read through the article, I couldn't help but notice that there may be some room for improvement, particularly in the presentation of figures, tables, and methods. I believe that enhancing these aspects could significantly enhance the overall quality of your work.

Allow me to make a suggestion that I believe will enhance the clarity and comprehensibility of your article. In the introduction and methods sections, I think it would greatly benefit the reader if you took the time to explicitly cite and define the components of "reporting" and "execution." By clearly outlining these terms, your audience will have a better understanding of their significance in relation to your study. The article would be improved if all the topics addressed within the discussion were presented in both the methodology and the results sections.

Moreover, to provide a more systematic and organized approach, I suggest considering the inclusion of a table that outlines each component, its definition, and the specific problem or solution associated with each step. This will serve as a valuable reference for readers, enabling them to navigate through your work with ease. The specific suggestions are in the attached text.

I appreciate the effort and dedication you have put into your research, and I genuinely believe that implementing these suggestions will make your work even more valuable to the academic community.

Thank you for considering my feedback. I look forward to seeing how your article progresses and the positive impact it will have on your readers.

Warm regards,

Reviewer #2: Dear authors and editors,

I have finished my review of the study "PONE-D-22-24320" which made a meta-evaluation of reviews addressing the quality of reporting and execution of meta-analyses on ecology. The authors concluded that the quality of reporting is improving but there is room for a higher improvement, especially on several characteristics of execution. I think that this study is interesting and comprehensive concerning the reviews and its studies used to describe the quality of meta-analysis conduction in ecology. I have only small suggestions for improvement, maybe the most important would be to include specific suggestions of what can be done to improve the "Compliance with reporting criteria" and "Non-independence". Furthermore, seems that the numbers of studies do not match at the end of the PRISMA flowchart (Figure 1).

Please, find details in the pdf file.

Kind regards.

PS: Please, consider submitting the main text with line numbering to ease referring to specific parts.

Reviewer #3: Dear authors, I your their approach simple, interesting, and quite refreshing, because of the broad look into the quality of meta-analyses in the field of Ecology. I also found the solutions proposed very interesting and with a real potential to improve the quality of future meta-analytical studies. I added some comments/suggestions that I hope may improve the quality of your manuscript. In addition, I do recommend that the manuscript be revised in terms of adherence to English language norms because I found several errors concerning primarily the usage of articles and prepositions.

Please number the lines in the text to facilitate review.

Please remove the citation from the abstract

Please review Figure 1.

The flowchart is an important component in systematic reviews, especially in one reviewing standardization and quality. I found several issues with the current flow chart. First, PRISMA’s flowchart was updated in 2019. I recommend the current version be used in this paper. In addition, the box “full-text articles excluded with reasons” does not depict the reasons which is important. Also, the number of articles in “All potential relevant papers assessed” is wrong for three reasons: (i) the number does not add up correctly, (ii) needs English review, and (iii) sich item is not part of the model Flowchart. Please review. I suggest consulting the following material http://www.prisma-statement.org/documents/EcoEvo_SI.pdf

In the manuscript you state “We compiled information from 19 papers (Table 1) that passed the inclusion criteria of having quantitative data on the quality of reporting (...)” while the flowchart shows 19 papers for the qualitative synthesis and 18 for the quantitative synthesis. Which is correct? Please review.

You removed Romanelli et al. [26] from the final comparison. So where did the 18 papers come from? I suggest following the reasoning that you removed duplicates when they occurred under similar analysis.

In the paragraph beginning with “In addition, because we had access to the full set of meta-analyses reviewed by Pappalardo et al. [10]” you refer to having expanded their results. Thus, do the procedures described in this paragraph all concern this complementation? Further on, does the subsequent paragraph also refer to the same complementation? If so, please highlight it in the second paragraph.

You have shown that some papers evaluated quality criteria that other didn’t. In such cases the “non-evaluation” of a criterion by a review paper does not imply that none of the meta-analyses evaluated met the criterion. Did you account for this (possible) source of bias (i.e., using ‘na’, reanalyzing papers, or statistically controlling such differences…)?

Fig. 2: Please specify “percent of papers complying with each REPORTING criterion ...”

Fig. 3: Please specify “percent of papers complying with each EXCECUTION criterion ...”

Did the review papers included in your study provide the scores assigned to each meta-analysis evaluated?

“Weighing” section of the discussion: I do understand with your statement that “weighting is not necessarily a requirement of a well-executed meta-analysis”. However, I urge caution in such statement. If published it may serve as argument for others conducting meta-analysis that simply do not want the trouble of searching for effect sizes, or do not comprehend the statement in depth.

You see… even with the many papers showing the importance of complying with both reporting and execution criteria there are still way too many papers missing important points. I truly believe we need a more mature academic context as meta-research is concerned before presenting such arguments without a solid evidence-based study accompanying it.

Please remove the statement or add a major “but” highlighting the caution when applying this reasoning.

Please change “A common and prominent issues in (…)” to “ Common and prominent issues in (…)”

Please change “This could reflect limitations imposed by software” to “This could reflect limitations imposed by the software”

Please change “or correlation-based approach” to “or correlation-based approaches”

Please change “the majority of” to “most”

Please make the following sentence clearer: “This conclusion is likely conservative because we were not strict in our criteria and did not require the authors referring to these analyses explicitly as “sensitivity analysis”.”

In Appendix A, please change “Matched to the results for” to “complied with the criterion” or another similar option because this term is quite hard to understand.

6. PLOS authors have the option to publish the peer review history of their article (what does this mean?). If published, this will include your full peer review and any attached files.

Reviewer #1: **Yes: **Fernanda Melo Carneiro

Reviewer #2: No

Reviewer #3: No

---

## [Author Response · Author response to Decision Letter 0]

14 Jul 2023

[Note – in the WORD file of this response, all our responses are embedded in blue italicized font and there is a figure included that may not show up here corresponds to new figure 4; line numbers in the response refer to the final (clean) manuscript version, not to the track-changes version]

https://journals.plos.org/plosone/s/file?id=ba62/PLOSOne_formatting_sample_title_authors_affiliations.pdf.]

--- Response: We did minor edits to the title page and manuscript to follow style guidelines. We also fixed the file names for the figures. Our supplementary material does not quite fit the format of the example so we did our best to match the naming style.

--- Response: We modified one reference, and added one reference for a website. Both were mentioned in the cover letter.

Additional Editor Comments:

Dear Dr. Pappalardo,

After this first review round your manuscript has received very favorable decisions. All three reviewers recommended a minor review should be implemented before your manuscript is accepted for publication in PLoS One Consider the extent of improvements, I will grant you a one-month period to deliver the new version of your manuscript to PLoS One. Please do not forget to prepare a rebuttal letter informing the authors what were the changes made and justifying those suggestions that were not accepted by the authors.

Congratulations on your hard work on this MS.

--- Response: Thank you very much for the good news and for your time in handling our manuscript.

Sincerely,

Daniel Silva

Response To Reviewers' comments:

Reviewer #1: Dear Authors, your article is quite interesting. The topic you chose is captivating, and your insights are thought-provoking.

However, as I read through the article, I couldn't help but notice that there may be some room for improvement, particularly in the presentation of figures, tables, and methods. I believe that enhancing these aspects could significantly enhance the overall quality of your work.

--- Response: We have completely remade the PRISMA plot in Figure 1. We have modified labels in Figure 2 and 3. We redid Table 2 following your advice.

Allow me to make a suggestion that I believe will enhance the clarity and comprehensibility of your article. In the introduction and methods sections, I think it would greatly benefit the reader if you took the time to explicitly cite and define the components of "reporting" and "execution." By clearly outlining these terms, your audience will have a better understanding of their significance in relation to your study. 

--- Response: Thank you for the suggestion. We have edited the text in the Introduction (lines 54-68) and Methods accordingly (lines 91-94). Table 2 lists all criteria included under 

Reporting and Execution and was modified following your suggestions.

The article would be improved if all the topics addressed within the discussion were presented in both the methodology and the results sections.

--- Response: We appreciate the suggestion, and added subtitles in the Methods to match the Results section. We also modified subtitles in the Discussion to match more closely the Methods and Results. 

Moreover, to provide a more systematic and organized approach, I suggest considering the inclusion of a table that outlines each component, its definition, and the specific problem or solution associated with each step. This will serve as a valuable reference for readers, enabling them to navigate through your work with ease. The specific suggestions are in the attached text.

--- Response: We have remade Table 2, which now lists each component and its definition in both the Reporting and Execution categories. As noted above, we also have expanded our definitions for “Reporting” and “Execution” in the main text. The problems and solutions associated with each criterion are detailed in the Discussion section (and now Table’s 2 caption points to Discussion for additional details). Following comments by Reviewer 2, we clarified the text related with problems and solutions and/or added text when they were not made explicit before.

I appreciate the effort and dedication you have put into your research, and I genuinely believe that implementing these suggestions will make your work even more valuable to the academic community.

--- Response: We are grateful for the time you spent reviewing our paper and for your helpful insights.

Thank you for considering my feedback. I look forward to seeing how your article progresses and the positive impact it will have on your readers.

Warm regards,

[Below we address Reviewer’s 1 comments that they included in the separate pdf file]

You haven't focus in the areas!

--- Response: We think our manuscript highlight the areas that need more work, such as addressing non-independence (including for phylogenetic non-independence), exploring temporal trends, multifactorial analysis of moderators. In terms of reporting, providing the full dataset and code (when applicable) to make analysis reproducible is one of the areas with low compliance. We have added text in the Discussion to be more specific about issues and solutions.

It will improve if you cite and define the components of the "reporting" and "execution" in the introduction and methods. I think that will be better if you put a table with the each one, definition and the problem associated or solution in specify the steps.

--- Response: We agree and appreciate the suggestion. We have edited the text in the Introduction (lines 54-68) and Methods accordingly (lines 91-94).The new version of Table 2 lists all criteria included under Reporting and Execution and was modified following your suggestions.

Reorganize the table as the content is confusingly adjusted.

--- Response: We have remade Table 2, which now lists each component and its definition in both the Reporting and Execution categories

Why did not all 18 articles analyzed appear in the figures?

--- Response: We double-checked and all the 18 articles appear in Figures 2 and 3 (depending on which criterion they provided data for).

Put the percentage of articles that consider each criterion analyzed.

--- Response: We were unclear on what the reviewer meant, so we could not address this comment.

Did you associate the use of dependence analysis with the articles? I think it would be clearer if you associated the presence of dependence analysis with a ratio between the number of papers and the number of effect sizes.

--- Response: We have modified the figure by identifying papers by the acknowledged source of non-independence. Many of the studies that have a large ratio of number of effect sizes to number of papers (i.e., residuals from the one-to-one line) do not address the non-independence. 

Perhaps the difference between areas within ecology? The types of questions or even the different study protocols, sometimes even for the same group of organisms.

--- Response: This is a good point. We added the following text: “…and to ecological studies often being more complex and varied in terms of the types of questions, sources of data, and experimental design”.

I think that this need to be present in the results with quantitative details.

--- Response: We took two approaches to address this comment: 1) When reporting in Results that compliance was highly variable for the criteria “quantifying heterogeneity in effect sizes”, we added additional detail: “ranging from 33% to 92% for weighting and 22% to 100% for quantifying heterogeneity”. In the Discussion, me modified the paragraph with the sentence you highlighted, now in lines 388-391.

I think that this aspect should be better discuted a long of the study to explain the different protocols adopted in the meta analysis.

--- Response: Thank you for the suggestion. We expanded on the most common methods used to address time lag in lines 434-439.

It is strange this appear just in the end of the discussion. The term confidence intervals was not introduce in any part of the text.

--- Response: Thank you for noticing this, that was also pointed out by Reviewer 2. We added text to develop the idea more fully and moved it up in the Discussion (now in lines 454-464).

Reviewer #2: Dear authors and editors,

I have finished my review of the study "PONE-D-22-24320" which made a meta-evaluation of reviews addressing the quality of reporting and execution of meta-analyses on ecology. The authors concluded that the quality of reporting is improving but there is room for a higher improvement, especially on several characteristics of execution. I think that this study is interesting and comprehensive concerning the reviews and its studies used to describe the quality of meta-analysis conduction in ecology. I have only small suggestions for improvement, maybe the most important would be to include specific suggestions of what can be done to improve the "Compliance with reporting criteria" and "Non-independence". 

--- Response: We appreciate and followed all your suggestions. Below, we provide specific answers to each comment. 

Furthermore, seems that the numbers of studies do not match at the end of the PRISMA flowchart (Figure 1).

--- Response: We provide detailed answers to your PRISMA comments below. We revised the PRISMA chart in the updated manuscript.

Please, find details in the pdf file.

Kind regards.

PS: Please, consider submitting the main text with line numbering to ease referring to specific parts.

--- Response: Done

[Below we address Reviewer’s 2 comments from the pdf]

Methods, 5th paragraph: It is not clear the difference between yes and partially. Note that authors definition allows an interpretation that both describe/cite one or more source of non-independence, and did not address all of them.

--- Response: We agree and have re-coded the two “partially” as “yes.” Thank you for pointing out this issue. 

Results, Non-independence, stretch "with some papers using a combination of these approaches": What was made in those cases? Were those studies counted twice or more times in Figure 5D?

--- Response: In our data collection, when two approaches were used, they were represented as for example, “choose, average”. Following that example for Figure 5D, that study would have contributed to both choose and average categories in the plot. We tried to make that explicit in the figure caption by saying “The papers that used more than one method (or source) were counted in each category, so the percentages between levels of panels B and D sum to greater than 100%.”

To make this clearer in the main text we added this example when referring to sources of non-independence “If a study reported both sources of non-independence, we recorded both (e.g., coding the study as “sample, study”)” 

1st paragraph of Discussion, "Even minimal [...] random-effects model).": What are author suggestion to correct these issues? It was nice that authors provide a suggestion below on data and code sharing, but it would be great to have some suggestions here too.

--- Response: Thank you for the suggestion. On lines 320-322, we have now added “To remedy this issue, we suggest academic journals adopt standard checklists for reporting items, such as the PRISMA-EcoEvo checklist [Table I, 2]”.

Discussion 1st paragraph "To encourage [...] code).": Some journals also cover for the fees from popular data repositories such as Dryad. It is important to highlight this incentive that already exist and is a good strategy from these journals.

--- Response: This is a good point and we agree it should be highlighted. Now it reads “To encourage code and data sharing, journals can develop incentives. Some cover the fee for publishing data in a repository. Discounts on open access fees could further encourage authors to share code and data.”

Discussion 1st paragraph "This is particularly [...] 52]": Why it is particularly important for these areas of research?

--- Response: Thank you for noticing this statement needed more context. We modified that sentence and the previous one as follows: “Achieving at least computational reproducibility will ensure results are robust, transparent, and credible. This is particularly important for researchers working in applied science and conservation where stakes are high, and transparency can help to maintain public trust”

Discussion 2nd paragraph last two sentences: Move to results.

--- Done. We agree that these details were not needed in the Discussion, and left only the following “We suggest that introducing the PRISMA Eco-Evo guidelines to a journal’s submission process will greatly benefit the discipline, especially if the journal publishes a large number of meta-analyses as is the case for Ecology Letters, Global Change Biology, Ecology, Oecologia and American Naturalist (the top five according to our compilation, Table S1).”. These results were not directly related to our main question and we think they fit best in the supporting information

Discussion Weighthing paragraph, "(and is one of the criteria in PRISMA-EcoEvo)": It is not clear this statement. Would it be more "precise", then "recommended by PRISMA-EcoEvo", or yielding precise estimates is a criteria of PRISMA-EcoEvo?

--- Response: Thank you for pointing this out. We have revised this section to increase clarity: see lines 364-366.

Discussion, Publication bias, 2nd paragraph: Consider to describe in brief these techniques.

--- Response: Thank you for the suggestion, the edited manuscript now includes a brief description of the techniques, see lines 434-439.

Discussion, "We recommend that readers consult Nakagawa et al. [22] for details of these methods.": Delete this sentence.

--- Response: We deleted it.

Discussion, Non-independence section: Solutions for non-independence in meta-analysis are lacking here. What authors recommend?

--- Response: We have added our recommendations (see lines 476-481 for sampling level, and 484-487 for study level).

Discussion, 1st paragraph of Non-independence: Writing here is difficult to follow. Consider editing to "Non-independence may occur at the sampling or at the study level, i.e., non-independent within-study error level or non-independent random effect, respectively".

--- Response: We revised the text as suggested.

Discussion, Non-independence section 2nd paragraph: Phylogenetic dependence may be challenging to address for groups that do not have a fully solved phylogenetic relationship (e.g. insects and aquatic macrophytes). What authors suggests for those cases?

--- Response: Very good point. We have addressed this point on Lines 505-508.

Discussion, How can we implement best practices? "Finally, [...] soon.": Consider referring only to resources that already are available or to provide the link to the website that will have this lectures. It is very speculative for us readers having only this future expectation of a forthcoming resource.

--- Response: We have added the citation and url for this site.

Discussion, How can we implement best practices? sentence "Based on the results [...] checklist.": This sentence seems to be misplaced. Consider removing it or developing the idea of this recommendation. For instance, what specific result from Pappalardo et al. [10] grounds this recommendation?

--- Response: Thank you for noticing this was a bit disconnected. We developed the idea more fully and moved it up in the Discussion (now in lines 454-464).

Discussion, How can we implement best practices? stretch "1) making [...] meta-analysis": Maybe "is not a statistically-focused"?

--- Response: We appreciate and followed your suggestion.

Supplementary Figures: Some supplementary figures are duplicated from the main text.

--- Response: We fixed this issue. We initially were going to include the PRISMA diagram in the supplement but communication with the journal indicated it should be in Figure 1 of the main text and we forgot to modify the R code to reflect this change.

Fig.1: Strange numbers in the last 4 boxes: All potential relevant papers (All; n = 205) > Excluded (E; 42) > Synthesis (S; 19). This would imply A–l - E = S: 2–5 - 42 = 19?s

--- Response: We appreciate that you double-checked our numbers. We updated the PRISMA plot. Here the source of the issue:

The 205 is a copy-paste error, we copied the R code to make the PRISMA plot from Pappalardo et al., 2020 code, and failed to edit the labels for that specific section (205 is the number of full articles assessed by Pappalardo et al. 2020).

The new PRISMA plot version clarifies the numbers of papers in each step, and we also added text in lines 110-115 to detail the issue with the two Romanelli papers. We also removed the excluded Romanelli paper from Table 1 which was probably adding to the confusion.

Reviewer #3: Dear authors, I your their approach simple, interesting, and quite refreshing, because of the broad look into the quality of meta-analyses in the field of Ecology. I also found the solutions proposed very interesting and with a real potential to improve the quality of future meta-analytical studies. I added some comments/suggestions that I hope may improve the quality of your manuscript. In addition, I do recommend that the manuscript be revised in terms of adherence to English language norms because I found several errors concerning primarily the usage of articles and prepositions.

--- Response: Dear reviewer, thank you for your time. Your comments have helped us improve the manuscript. 

Please number the lines in the text to facilitate review.

--- Response: Done. 

Please remove the citation from the abstract

--- Response: We changed the format to avoid the passive citation of O’Dea et al., but we retained the citations. We think the citation of Pappalardo et al. is necessary, and we wanted to mention O’Dea et al to avoid a misleading impression that our study developed the PRISMA EcoEvo checklist.

Please review Figure 1.

The flowchart is an important component in systematic reviews, especially in one reviewing standardization and quality. I found several issues with the current flow chart. First, PRISMA’s flowchart was updated in 2019. I recommend the current version be used in this paper. In addition, the box “full-text articles excluded with reasons” does not depict the reasons which is important. Also, the number of articles in “All potential relevant papers assessed” is wrong for three reasons: (i) the number does not add up correctly, (ii) needs English review, and (iii) sich item is not part of the model Flowchart. Please review. I suggest consulting the following material http://www.prisma-statement.org/documents/EcoEvo_SI.pdf

--- Response: We completely redid the PRISMA plot in Figure 1: (i) We corrected the numbers. The 205 is a copy-paste error, we copied the R code to make the PRISMA plot from Pappalardo et al., 2020 code, and failed to edit the labels for that specific section (205 is the number of full articles assessed by Pappalardo et al. 2020). The 19/18 confusion is because we initially compiled the information for 19 papers and then realized the same dataset was likely used in two papers and excluded 1 paper to get the final 18. Now we included that in the other “exclusion” stage. We also added text on lines 110-115 to explain the case with the two Romanelli papers in detail. We also clarified that there are 19 cases because Lodi et al. provided a review of meta-analyses in two separate study areas (so that paper contributed two cases). We also added the different reasons for exclusion and the number of papers excluded by each reason. (ii) We switched to text suggested by O’Dea et al, 2021 for the main boxes. (iii) We followed the classic PRISMA plot model suggested in Fig. 3A in O’Dea et al, 2021 (PRISMA EcoEvo).

In the manuscript you state “We compiled information from 19 papers (Table 1) that passed the inclusion criteria of having quantitative data on the quality of reporting (...)” while the flowchart shows 19 papers for the qualitative synthesis and 18 for the quantitative synthesis. Which is correct? Please review.

--- Response: We have now clarified this point on lines 110-115. The two Romanelli papers are complementary analyses of the same dataset, so we removed one from the final analyses and we did not present the excluded Romanelli paper in Table1. 

You removed Romanelli et al. [26] from the final comparison. So where did the 18 papers come from? I suggest following the reasoning that you removed duplicates when they occurred under similar analysis.

--- Response: Please see the preceding comment (and lines 110-115, the revised Table 1, and new PRISMA plot figure)

In the paragraph beginning with “In addition, because we had access to the full set of meta-analyses reviewed by Pappalardo et al. [10]” you refer to having expanded their results. Thus, do the procedures described in this paragraph all concern this complementation? 

--- Response: Yes, this entire paragraph is about new data collected from the same meta-analyses analyzed by Pappalardo et al [10]. We made edits on lines 164-173 to clarify these distinctions, we also identified the new information in Table 2 and in Appendix S1. 

Further on, does the subsequent paragraph also refer to the same complementation? If so, please highlight it in the second paragraph.

--- Response: Yes, it does. We added at the beginning of the paragraph: “A portion of the new data we collected from the meta-analyses reviewed by Pappalardo et al. [1] focused on non-independence” to clarify this issue.

You have shown that some papers evaluated quality criteria that other didn’t. In such cases the “non-evaluation” of a criterion by a review paper does not imply that none of the meta-analyses evaluated met the criterion. Did you account for this (possible) source of bias (i.e., using ‘na’, reanalyzing papers, or statistically controlling such differences…)?

--- Response: We do not think there is bias in data from each review paper, because when a review paper assessed a specific criterion, all the meta-analyses included in the review were coded. And the percentage of compliance for each review is calculated (or was calculated in the original publication) based on all of the meta-analysis coded.

In terms of comparing compliance across reviews, we are not calculating an aggregate measure of compliance across review papers, or making any statistical inference across review papers. The comparison is qualitative to help us summarize the items where we are doing well and which items that require more attention. 

Fig. 2: Please specify “percent of papers complying with each REPORTING criterion ...”

--- Response: Added, thank you for the suggestion.

Fig. 3: Please specify “percent of papers complying with each EXCECUTION criterion ...”

--- Response: Added, thank you for the suggestion.

Did the review papers included in your study provide the scores assigned to each meta-analysis evaluated?

--- Response: Some papers did calculate the percentage of papers complying. Others reported the number of papers complying. Others presented raw data that allowed us to calculate the percentage complying. Appendix S1 describes the specific criteria from each review, how we matched it to our list of criteria, how the original review reported the measure of interest, and if needed, how we converted it to percent compliance. The Notes column in the “paperTopics” table in the data file “compilation-of-previous-review-papers” also has details on the data source.

“Weighing” section of the discussion: I do understand with your statement that “weighting is not necessarily a requirement of a well-executed meta-analysis”. However, I urge caution in such statement. If published it may serve as argument for others conducting meta-analysis that simply do not want the trouble of searching for effect sizes, or do not comprehend the statement in depth.

You see… even with the many papers showing the importance of complying with both reporting and execution criteria there are still way too many papers missing important points. I truly believe we need a more mature academic context as meta-research is concerned before presenting such arguments without a solid evidence-based study accompanying it.

Please remove the statement or add a major “but” highlighting the caution when applying this reasoning.

--- Response: PP agrees with the concern about this statement and removed it. This was a source of healthy academic debate between the co-authors and the text went back and forward during revisions. Based on a comment by Reviewer #2, we added this sentence to clarify recommendations “The PRISMA-EcoEvo guidelines recommend using a weighted analysis because weighting generally yields more precise estimates of effects that unweighted analyses [2].”

Please change “A common and prominent issues in (…)” to “ Common and prominent issues in (…)”

--- Response: Thank you for pointing out a problem here. It should be singular, because we are mentioning only one issue. We changed it to “A common and prominent issue in exploring heterogeneity is…”

Please change “This could reflect limitations imposed by software” to “This could reflect limitations imposed by the software”

--- Response: We changed it as suggested, thank you.

Please change “or correlation-based approach” to “or correlation-based approaches”

--- Response: We changed it as suggested, thank you.

Please change “the majority of” to “most”

--- Response: We changed it as suggested, thank you.

Please make the following sentence clearer: “This conclusion is likely conservative because we were not strict in our criteria and did not require the authors referring to these analyses explicitly as “sensitivity analysis”.”

--- Response: We removed the sentence since it was not adding much, and Appendix S1 details how we coded studies for the criterion “sensitivity analysis”. 

In Appendix A, please change “Matched to the results for” to “complied with the criterion” or another similar option because this term is quite hard to understand.

--- Response: We revised the Appendix and switched all the cases where that phrase appeared to “Data extracted from criterion..”

---

## [Decision Letter · Decision Letter 1]

12 Sep 2023

PONE-D-22-24320R1A meta-evaluation of the quality of reporting and execution in ecological meta-analysesPLOS ONE

Dear Dr. Pappalardo,

Thank you for submitting your manuscript to PLOS ONE. After careful consideration, we feel that it has merit but does not fully meet PLOS ONE’s publication criteria as it currently stands. Therefore, we invite you to submit a revised version of the manuscript that addresses the points raised during the review process.

We look forward to receiving your revised manuscript.

Kind regards,

Daniel de Paiva Silva, Ph.D.

Academic Editor

PLOS ONE

Journal Requirements:

Reviewers' comments:

Reviewer's Responses to Questions

**Comments to the Author**

1. If the authors have adequately addressed your comments raised in a previous round of review and you feel that this manuscript is now acceptable for publication, you may indicate that here to bypass the “Comments to the Author” section, enter your conflict of interest statement in the “Confidential to Editor” section, and submit your "Accept" recommendation.

Reviewer #2: All comments have been addressed

Reviewer #3: All comments have been addressed

2. Is the manuscript technically sound, and do the data support the conclusions?

Reviewer #2: Yes

Reviewer #3: Yes

3. Has the statistical analysis been performed appropriately and rigorously? 

Reviewer #2: N/A

Reviewer #3: Yes

4. Have the authors made all data underlying the findings in their manuscript fully available?

Reviewer #2: Yes

Reviewer #3: Yes

5. Is the manuscript presented in an intelligible fashion and written in standard English?

Reviewer #2: Yes

Reviewer #3: Yes

6. Review Comments to the Author

Reviewer #2: Dear authors and Editor,

I think that the reviewed version of the manuscript “A meta-evaluation of the quality of reporting and execution in ecological meta-analyses” included and/or rebutted reasonably all the points observed by all reviewers. I read only small points (most are typos):

l. 107: "PDFs of".

l. 114: use ";" instead of ":".

l. 129-130: "Romanelli et al. Reviews [14, 15]".

l. 205: "[...] across species);".

l. 277-278: I understood the message here, but in my opinion, the example here should be "e.g., multiple effect sizes per publication". I will extend this minor point because there are not many corrections to indicate in this study... I (gladly) ended more thinking than suggesting corrections here again.

Several independent experiments can be published in a single paper. There are prospective meta-analyses where a set of independent experiments are conducted to be combined with meta-analytical tools (Ioannidis 2017. DOI: 10.1136/bjsports-2017-097621; e.g. Kawaguchi 2013. DOI: 10.1264/jsme2.me13014). This would not be an "explicit" violation of the independence of effect sizes (I am thinking here of effect size multiplicity only; Lopez-Lopez et al. 2018. DOI: 10.1002/jrsm.1310). Sure, we can think of the possibility of effect sizes being published by the same research group as being more similar than those from different research groups, but this would be a dependence above the study level.

l. 351: "PRISMA".

l. 361-362: Not needed. L. 360-361 could be joined to l. 363.

l. 549: "framework),".

Congratulations to the authors for conducting this study! It was a pleasure to read it. Looking forward to the published version to recommend it in classes and citing.

Best regards,

Jean C. G. Ortega

Reviewer #3: Thank you for promptly considering the suggestions made to improve your manuscript. Find below some other small details, mostly associated with language.

L23: Each of the 18 review papers

L29: … a wide variation

L63-65: Sentence is confusing. I suggest changing to:

“The quality of a meta-analysis also is affected by how well the study is implemented. We refer to this proper implementation as execution quality, which is the extent to which the analyses conform to expert recommendation.”

L111: change “(…) in restoration ecology; in [14] (...)” to “Romanelli et al. [14] describe (...)”

L116: Meta-analysis should be plural

L117: separate should be singular

L159: provided should be in the past tense

L217-218: change “is provided” to “are provided”

L377: change : to ,

L410: remove “as” in “publication bias can be considered as a type”

L354-455: use “analysis concerned the method…” or “analysis was associated with”

7. PLOS authors have the option to publish the peer review history of their article (what does this mean?). If published, this will include your full peer review and any attached files.

Reviewer #2: **Yes: **Jean Carlo Gonçalves Ortega

Reviewer #3: No

---

## [Author Response · Author response to Decision Letter 1]

22 Sep 2023

[Note – all our responses are embedded here in blue italicized font; line numbers in the response refer to the final (clean) manuscript version, not to the track-changes version]

Journal Requirements:

--- Response: We have reviewed the references, fixed the error on the text found by Reviewer #2, and made no changes to the overall list of references. 

Response To Reviewers' comments:

Reviewer #2:

Dear authors and Editor,

I think that the reviewed version of the manuscript “A meta-evaluation of the quality of reporting and execution in ecological meta-analyses” included and/or rebutted reasonably all the points observed by all reviewers. I read only small points (most are typos):

l. 107: "PDFs of".

--- Response: Fixed. Thank you for catching this typo.

l. 114: use ";" instead of ":".

--- Response: Done.

l. 129-130: "Romanelli et al. Reviews [14, 15]".

--- Response: Fixed, thank you for catching this.

l. 205: "[...] across species);".

--- Response: Parenthesis added, thank you for catching this.

l. 277-278: I understood the message here, but in my opinion, the example here should be "e.g., multiple effect sizes per publication". I will extend this minor point because there are not many corrections to indicate in this study... I (gladly) ended more thinking than suggesting corrections here again.

Several independent experiments can be published in a single paper. There are prospective meta-analyses where a set of independent experiments are conducted to be combined with meta-analytical tools (Ioannidis 2017. DOI: 10.1136/bjsports-2017-097621; e.g. Kawaguchi 2013. DOI: 10.1264/jsme2.me13014). This would not be an "explicit" violation of the independence of effect sizes (I am thinking here of effect size multiplicity only; Lopez-Lopez et al. 2018. DOI: 10.1002/jrsm.1310). Sure, we can think of the possibility of effect sizes being published by the same research group as being more similar than those from different research groups, but this would be a dependence above the study level.

--- Response: This is a good point, we edited the text as suggested.

l. 351: "PRISMA".

--- Response: Fixed. Thank you for catching this typo.

l. 361-362: Not needed. L. 360-361 could be joined to l. 363.

l. 549: "framework),".

--- Response: Edited as suggested.

Congratulations to the authors for conducting this study! It was a pleasure to read it. Looking forward to the published version to recommend it in classes and citing.

--- Response: We really appreciate all your time to re-review the manuscript and all the suggestions to make it better.

Best regards,

Jean C. G. Ortega

Reviewer #3:

Thank you for promptly considering the suggestions made to improve your manuscript. Find below some other small details, mostly associated with language.

--- Response: Thank you very much for all the suggestions and your time to re-review and help us improve our manuscript.

L23: Each of the 18 review papers

--- Response: Edited as suggested.

L29: … a wide variation

--- Response: Edited as suggested.

L63-65: Sentence is confusing. I suggest changing to:

“The quality of a meta-analysis also is affected by how well the study is implemented. We refer to this proper implementation as execution quality, which is the extent to which the analyses conform to expert recommendation.”

--- Response: Edited as suggested.

L111: change “(…) in restoration ecology; in [14] (...)” to “Romanelli et al. [14] describe (...)”

--- Response: Edited as suggested.

L116: Meta-analysis should be plural

--- Response: Edited as suggested.

L117: separate should be singular

--- Response: Edited as suggested.

L159: provided should be in the past tense

--- Response: Edited as suggested.

L217-218: change “is provided” to “are provided”

--- Response: Edited as suggested.

L377: change : to ,

--- Response: Edited as suggested.

L410: remove “as” in “publication bias can be considered as a type”

--- Response: Edited as suggested.

L354-455: use “analysis concerned the method…” or “analysis was associated with”

--- Response: Edited as the second option.

---

## [Editor Report · Decision Letter 2]

25 Sep 2023

A meta-evaluation of the quality of reporting and execution in ecological meta-analyses

PONE-D-22-24320R2

Dear Dr. Pappalardo,

We’re pleased to inform you that your manuscript has been judged scientifically suitable for publication and will be formally accepted for publication once it meets all outstanding technical requirements.

Kind regards,

Daniel de Paiva Silva, Ph.D.

Academic Editor

PLOS ONE

Additional Editor Comments (optional):

Dear Dr. Papparlardo,

I am pleased to accept your manuscript fopr publication in PLoS One! Congratulations on your hard work!

Best regards,

Daniel Silva, PhD
---

## [Editor Report · Acceptance letter]

2 Oct 2023

PONE-D-22-24320R2 

A meta-evaluation of the quality of reporting and execution in ecological meta-analyses 

Dear Dr. Pappalardo:

I'm pleased to inform you that your manuscript has been deemed suitable for publication in PLOS ONE. Congratulations! Your manuscript is now with our production department. 

Kind regards, 

on behalf of

Dr. Daniel de Paiva Silva 

Academic Editor

PLOS ONE